# Differentiation of *Lacticaseibacillus zeae* Using Pan-Genome Analysis and Real-Time PCR Method Targeting a Unique Gene

**DOI:** 10.3390/foods10092112

**Published:** 2021-09-07

**Authors:** Eiseul Kim, Seung-Min Yang, Hae-Yeong Kim

**Affiliations:** Department of Food Science and Biotechnology, Institute of Life Sciences & Resources, Kyung Hee University, Yongin 17104, Korea; eskim89@khu.ac.kr (E.K.); ysm9284@gmail.com (S.-M.Y.)

**Keywords:** *Lacticaseibacillus zeae*, *Lacticaseibacillus* species, real-time PCR, pan-genome, unique gene, identification, fermented dairy product

## Abstract

*Lacticaseibacillus zeae* strains, isolated from raw milk and fermented dairy products, are closely related to the *Lacticaseibacillus* species that has beneficial probiotic properties. However, it is difficult to distinguish those using conventional methods. In this study, a unique gene was revealed to differentiate *L. zeae* from other strains of the *Lacticaseibacillus* species and other species by pan-genome analysis, and a real-time PCR method was developed to rapidly and accurately detect the unique gene. The genome analysis of 141 genomes yielded an 17,978 pan-genome. Among them, 18 accessory genes were specifically present in five genomes of *L. zeae*. The glycosyltransferase family 8 was identified as a unique gene present only in *L. zeae* and not in 136 other genomes. A primer designed from the unique gene accurately distinguished *L. zeae* in pure and mixed DNA and successfully constructed the criterion for the quantified standard curve in real-time PCR. The real-time PCR method was applied to 61 strains containing other *Lacticaseibacillus* species and distinguished *L. zeae* with 100% accuracy. Also, the real-time PCR method was proven to be superior to the 16S rRNA gene method in the identification of *L. zeae*.

## 1. Introduction

*Lactobacillus* genus has been reclassified, as such species previously belonging to *Lactobacillus casei* group now are allotted to *Lacticaseibacillus* genus [1,2]. The genus *Lacticaseibacillus* consists of 26 species (*L. absianus*, *L. baoqingensis*, *L. brantae*, *L. camelliae*, *L. casei*, *L. chiayiensis*, *L. daqingensis*, *L. hegangensis*, *L. hulanensis*, *L. jixianensis*, *L. manihotivorans*, *L. mingshuiensis*, *L. nasuensis*, *L. pantheris*, *L. paracasei*, *L. porcinae*, *L. rhamnosus*, *L. saniviri*, *L. sharpeae*, *L. songhuajiangensis*, *L. suibinensis*, *L. suilingensis*, *L. thailandensis*, *L. yichunensis*, *L. zeae*, and *L. zhaodongensis*), and *Lacticaseibacillus zeae* is one of the members of the *Lacticaseibacillus* genus, along with *L. casei*, *L. paracasei*, *L. rhamnosus*, and *L. chiayiensis*. However, the taxonomic position of *L. zeae* has long been debated. In 2008, the use of the name *L. zeae* rejected for contravening Rules 51b (1) and (2) of the International Code of Nomenclature of Bacteria [3], and only the three species *L. casei*, *L. paracasei*, and *L. rhamnosus* were included in the *Lacticaseibacillus* species [4]. However, the name *L. zeae* has since been reported to be legitimate and was validly published [5]. *L. zeae* has also been justified as a designation for an independent species based on the results of phenotypic characterization and whole-genome sequence-based analysis [5]. With the recent revival of the name *L. zeae*, therefore, an accurate method is needed to detect this species.

Traditionally, lactic acid bacteria have been identified by biochemical analysis, but classical identification tools cannot distinguish among some species with similar phenotypes [6]. Therefore, molecular methodologies such as amplified ribosomal DNA restriction analysis (ARDRA), randomly amplified polymorphic DNA (RAPD), and repetitive sequence-based PCR have been used to identify lactic acid bacteria [7,8,9]. Among these methodologies, polymerase chain reaction (PCR) is more cost-effective and faster than other molecular tools for the identification of lactic acid bacteria [10]. The main standard marker for differentiation of lactic acid bacteria is the 16S rRNA gene, but it is difficult to discriminate among closely related species such as *Lacticaseibacillus* species using this marker [11,12,13]. In particular, this gene has a high sequence similarity between *L. zeae* and other *Lacticaseibacillus* species of 98.7–99.9%, so it cannot be used to accurately distinguish species in the group [14]. Therefore, an alternative novel target gene is needed as a marker for the identification of *L. zeae*.

Although it is possible to identify and differentiate lactic acid bacteria by whole genome sequencing, it is time-consuming and costly compared to molecular methodologies [15]. Recently, some researchers have developed a PCR method that can efficiently differentiate closely related bacterial species based on the whole genome analysis [12,13]. However, the development of the PCR method to distinguish *L. zeae* from other closely related species using a marker obtained based on the pangenome has rarely been reported. This study revealed a unique gene of *L. zeae* that can be used to accurately distinguish it from other *Lactobacillus*-related species based on pan-genome analysis, and a real-time PCR method was developed that can detect this unique gene by a designed primer.

## 2. Materials and Methods

### 2.1. Pan-Genome Analysis

A total of 141 genome sequences representing nine lactic acid bacterial species were downloaded from the National Center for Biotechnology Information (NCBI) (Table 1). To overcome the limitation that 141 genomes contained only 5 out 26 species in the genus *Lacticaseibacillus*, nine species isolated from raw milk, the main habitat of *L. zeae*, were included in the genome analysis [16]. The pan-genome was analyzed by a pan-genome workflow using the Anvi’o program version 6.0 [17]. The genome sequences were arranged based on the distribution of orthologous gene clusters using the Markov Cluster Algorithm (MCL). Pan-genome profiles of the *Lactobacillus*-related species genome sequences were generated using the bacterial pan-genome analysis pipeline (BPGA) as described in the manual provided by developers [18]. The protein files of 141 genome sequences obtained from NCBI served as the input file for the BPGA analysis. Protein homologs were then clustered by USEARCH with 50% sequence similarity as a cut-off, which is the default setting value. The pan- and core-genome phylogeny analyses were constructed using 20 random orthologous protein clusters [19]. Each orthologous cluster is aligned with a cluster of orthologous groups (COG) database (http://www.ncbi.nlm.nih.gov/COG/ accessed on 4 December 2020) to assign categories to representative protein sequences. Since some proteins in lactobacilli genomes can fit more than one COG classification, and some proteins have no COG assigned, COG analysis restricted the analysis to known protein types. The unique genes of *L. zeae* were discovered by analyzing the accessory-genome, the set of proteins present in some, but not all genomes, and *L. zeae* specific primer was developed by selecting a gene suitable for primer design among them.

The unique gene for *L. zeae* was compared with other strains through BLASTP search against the Kyoto Encyclopedia of Genes and Genomes (KEGG) database, and sequence alignment was performed using Clustal Omega (https://www.ebi.ac.uk/Tools/msa/clustalo/ accessed on 31 August 2021).

### 2.2. Bacterial Strains and DNA Extraction

The bacterial strains used in this study are listed in Table 2. All reference strains were collected from the Korean Culture Center of Microorganisms (KCCM, Seoul, Korea), the NITE Biological Resource Center (NBRC, Chiba, Japan), the Korean Collection for Type Cultures (KCTC, Daejeon, Korea), the Korean Agricultural Culture Collection (KACC, Jeonju, Korea), and the Microorganism and Gen Bank (MGB, Gwangju, Korea).

The isolated strain was isolated from raw milk. Raw milk sample was obtained from the ranch in Korea. The udder was washed prior to collecting raw milk and then directly placed into sterile tubes. After collection, raw milk was maintained at 4 °C during transfer to the laboratory. For isolation of *L. zeae*, the serially diluted samples were spread on lactobacilli MRS agar (Difco, Becton & Dickinson, Sparks, MD, USA) plate and incubated at 37 °C for 48 h under anaerobic conditions. The different colonies were selected and identified by 16S rRNA gene sequencing. An isolate suspected of *L. zeae* was selected and designated as the Laboratory Isolate (LI) 220.

All reference strains and isolate were cultured anaerobically in lactobacilli MRS broth (Difco) at 37 °C for 48 h. For extraction of genomic DNA, 1 mL of cultured cells was pelleted by centrifugation at 13,600× *g* for 10 min and suspended in 200 µL of lysis buffer. According to the manufacturer’s instruction, genomic DNA was extracted using a DNeasy Blood & Tissue kit (Qiagen, Hilden, Germany). DNA concentration and purity of reference strains were measured using a MaestroNano^®^ spectrophotometer (Maestrogen, Las Vegas, NV, USA).

### 2.3. Real-Time PCR Conditions

The specificity of the designed primer was confirmed using pure and mixed DNA of nine species, including species mainly found in raw milk and closely related to *L. zeae*. For the preparation of mixed DNA, DNA was extracted from the cells of nine species and 20 ng of each mixed to provide a template for PCR amplification. The standard curve for quantification was generated by *L. zeae* KACC 11461 serially diluted from 10^3^ to 10^9^ CFU/mL [20]. PCR was performed with CFX96 Deep Well Real-time System (Bio-Rad, Hercules, CA, USA) with a 20 µL reaction mixture containing 20 ng of DNA, 500 nM of primer pair, and 10 µL of 2X Thunderbird SYBR^®^ qPCR Mix (Toyobo, Osaka, Japan). The amplification consisted of an initial denaturation at 95 °C for 2 min, followed by 35 cycles of 95 °C for 5 s and 60 °C for 30 s. The amplicon was then heated to a temperature from 65 °C to 95 °C, with increments of 0.5 °C, to generate a melting curve.

### 2.4. Evaluation and Application of Real-Time PCR

The real-time PCR developed in this study was evaluated using 61 bacterial strains (Table 2). The real-time PCR was conducted according to the conditions described in the previous Section 2.3. The strain amplified by real-time PCR was verified by 16S rRNA gene sequencing with 27F/1492R primers. The amplicons were purified using the QIAquick PCR purification kit (Qiagen), and purified amplicons were sequenced. The 16S rRNA gene sequences were then confirmed by BLAST searches.

Our method was applied to spiked food sample. The milk sample was purchased from a local market in Korea and was confirmed to be free of *L. zeae*. A spiked milk sample was prepared according to a previous study [21,22]. Briefly, 25 mL milk sample was inoculated with *L. zeae* at a concentration of 10^3^ to 10^9^ CFU/mL. Genomic DNA was extracted inoculated sample according to the method described above, and real-time PCR was performed.

## 3. Results and Discussion

### 3.1. Pan-Genome Analysis

Previous studies have reported that public databases contain misnamed genomes for phenotypically closely related species [23,24,25]. Therefore, phylogenetic analysis based on pan-genome was performed to avoid causing incorrect results due to the misclassification of genome sequences used in this study prior to the genome analysis of *L. zeae*. The phylogenetic tree based on the pan-genome was mostly clustered by species, resulting in two major clusters (Figure 1). The first cluster included *L. helveticus* and *L. delbrueckii*, and the second included *Lacticaseibacillus* species, *L. plantarum*, and *L. brevis*. However, eight genomes of *L. casei* were clustered with *L. paracasei* instead of *L. casei*. Consistent with the results of a previous study [25], we suggested that *L. casei* 12A, 12/1, A2-362, UW4, Z11, NBRC 101979, UW1, and CRF28 should be moved out and placed into the *L. paracasei*. Similar results were also confirmed in the phylogenetic analysis based on binary pan-matrix and concatenated core genes (Figure 2). In the phylogenetic analysis, *L. casei*, *L. chiayiensis*, and *L. zeae* were very similar, which is similar to a previous study that three species were clustered adjacent to each other using a phylogenetic tree based on core genome MLST [5]. Therefore, it was confirmed that *L. casei* and *L. zeae* were differentiated by pan-genome analysis based on whole-genome sequences.

The 17,978 pan-genome obtained from 141 genomes is composed of 144 core genes and 4271 unique genes. The core, accessory, and unique gene clusters were further annotated into COG categories. The core-genome was mostly preserved in the following: transcription, ribosomal structure & biogenesis (38.6%), and nucleotide transport & metabolism (6.7%). Also, the genes common to five genome sequences of *L. zeae* were classified to COG categories, mainly functioning in the defense system (16.7%), amino acid transport and metabolism (11.1%), and cell wall/membrane/envelop biogenesis (11.1%). Among the 13,563 accessory genes, there were 18 genes common to five *L. zeae* genome sequences (94.0–100% sequence identity). The 18 genes were aligned with 72,899,005 bacterial sequences through blast analysis. As a result, three genes existed in other microorganisms such as *Enterococcus durans*, *Pediococcus damnosus*, and *P. acidilactici*, and the remaining 15 genes were specifically present in *L. zeae*. Only 15 genes presented in five genome sequences in *L. zeae*. Among these, the gene specific to *L. zeae* was finally selected as glycosyltransferase family 8 (accession no. KRK10099.1) in consideration of the GC content and length. Glycosyltransferase was also present in other lactic acid bacteria. The glycosyltransferase of *L. zeae* was compared with other bacterial strains, and as a result, it showed the highest homology with *Enterococcus gallinarum* (36% identity) and less than 34% homology with other *Lacticaseibacillus* strains (Appendix A). This gene was proven to be a gene specific to *L. zeae* because of its low sequence similarity of less than 36% with other species.

Glycosyltransferases are associated with bacterial stress response, biofilm formation, and sucrose metabolism [26]. This enzyme is also associated with exopolysaccharides (EPS) biosynthesis, which can be part of an important process related to probiotic characteristics such as auto-aggregation, colonization, and survival [27]. In the classification system, glycosyltransferases are divided into 111 families according to their amino acid sequences and differ in function and structure based on the family type (https://www.cazy.org accessed on 4 December 2020). A previous study has reported that lactobacilli encoded various families of glycosyltransferases that have different sequences depending on the species or strains [28]. The same family of glycosyltransferase can also have different sequences with diverse functions depending on their evolutionary origin or acquisition [29,30]. The glycosyltransferase family 8 (accession no. KRK10099.1) found in this study was conserved in genomes of *L. zeae* with high amino acid sequence similarity (>99%), whereas it showed low similarity in other species. As shown in Appendix A, since the sequences did not match consecutively, designing a primer at any position within this gene does not result in amplification in other bacteria. Therefore, we confirmed that this gene was specifically present in *L. zeae*.

### 3.2. Specificity Test

PCR is a well-known and powerful tool to accurately and rapidly detect lactic acid bacteria [31]. The accuracy of PCR depends on the specificity and sensitivity of the gene or primer used in the experiment. Previous studies have reported differentiating *L. zeae* using the 16S rRNA gene sequence and housekeeping genes such as *yycH* and *dnaK* gene as PCR markers [6,32]. However, these genes have high sequence similarities (about 80–100%) among other lactic acid bacterial strains and require an additional sequencing process that is costly and time-consuming. Therefore, this study developed a PCR method that can rapidly and accurately detect *L. zeae* by targeting a novel unique gene obtained from the pan-genome analysis.

The *L. zeae* specific primer was designed from the glycosyltransferase family 8 gene obtained from the pan-genome analysis (Table 3). The specificity of the primer was performed using pure and mixed DNA of nine species of *L. zeae*, *L*. *casei*, *L*. *paracasei*, *L*. *chiayiensis*, *L. rhamnosus*, *L*. *helveticus*, *L*. *plantarum*, *L*. *delbrueckii*, and *L. brevis*. *L. zeae* KACC 11461 presented a Ct value of 13.84 (Figure 3A), and amplicon presented Tm of 81 °C (Figure 3B). Other pure cultured strains did not show amplification for real-time PCR; and the mixed DNA of nine species was amplified only in a well containing *L. zeae* with a Ct value of 14.88 (Figure 3C), and amplicon presented Tm of 81 °C (Figure 3D). Other mixed cultures did not produce any amplification curve. Therefore, our method successfully amplified the glycosyltransferase family 8 gene in pure and mixed cultures of nine species, suggesting the possibility of identification of *L. zeae* in complex microbial samples.

The standard curve for *L. zeae* was constructed using serial dilutions in the range from 10^3^ to 10^9^ CFU/mL per reaction. The coefficients of correlation (*R*^2^) were 0.997, and amplification efficiency was 92.0%. The standard curve had a slope of −3.530 (Figure 4). A previous study reported that a standard curve with an *R*^2^ value ≥ 0.98 and slope value in the range of −3.1 to −3.6 is a high-efficiency real-time PCR assay [33]. Therefore, our real-time PCR method is considered a highly efficient method for the identification of *L. zeae*.

### 3.3. Evaluation of Real-Time PCR Developed in This Study

Real-time PCR was performed using 61 bacterial strains to evaluate whether the designed primer could exclusively detect *L. zeae*. *L. zeae* KACC 11461 and LI220 presented Ct values of 16.31 and 17.09, respectively, and all amplicon presented Tm of 81 °C (Figure 5). Other bacterial strains did not show amplification for real-time PCR, demonstrating 100% specificity. The amplified *L. zeae* strains were confirmed using the conventional identification method of 16S rRNA gene sequencing. 16S rRNA gene sequencing presented three different candidates: *L. casei*, *L. zeae*, and *L. rhamnosus*, instead of providing one species (Table 4). This is consistent with previous studies that reported closely related species were difficult to distinguish using 16S rRNA gene sequences due to the sequence similarity [13,34]. Therefore, it was shown that the real-time PCR developed in this study more accurately distinguished *L. zeae* than 16S rRNA gene sequencing, which is mainly used for microbial identification. Because this species is rare in the environment and food, it was difficult in this study to find an isolate and only a limited number of isolates could be used for PCR analysis. Also, our data has a shortcoming in that a low number of *Lacticaseibacillus* species (5 out of 26 species) represented in this study because species that have been described very recently have not been easily accessible isolates. However, since the study used a primer designed with genes analyzed using most of the available genomes, specificity and accuracy could be proven.

The quantification of genomic DNA in a food sample was conducted by artificially adding *L. zeae* strain to milk. The real-time PCR method developed in this study could successfully be used to identify *L. zeae* at a concentration of 10^3^ to 10^9^ CFU/mL in milk (Figure 6). Samples artificially inoculated with *L. zeae* (Ct values: 14.97 to 34.78) had a slightly higher Ct value than the pure culture of *L. zeae* strain, which seemed to slightly affect the efficiency of real-time PCR. This may be due to the presence of PCR inhibitors from fast and protein in food [21]. Therefore, our real-time PCR method was able to identify *L. zeae* in the food matrix.

## 4. Conclusions

In this study, the glycosyltransferase family 8 gene was revealed as a unique gene of *L. zeae* using a pan-genome analysis. The primer targeting the glycosyltransferase family 8 gene showed high specificity for 61 bacterial strains and was able to rapidly and efficiently distinguish and quantify *L. zeae*. It also showed higher accuracy than conventional identification methods targeting 16S rRNA gene sequences. Therefore, this method could be further applied to screen *L. zeae* in complex microbial communities in food samples.

## Figures and Tables

**Figure 1 foods-10-02112-f001:**
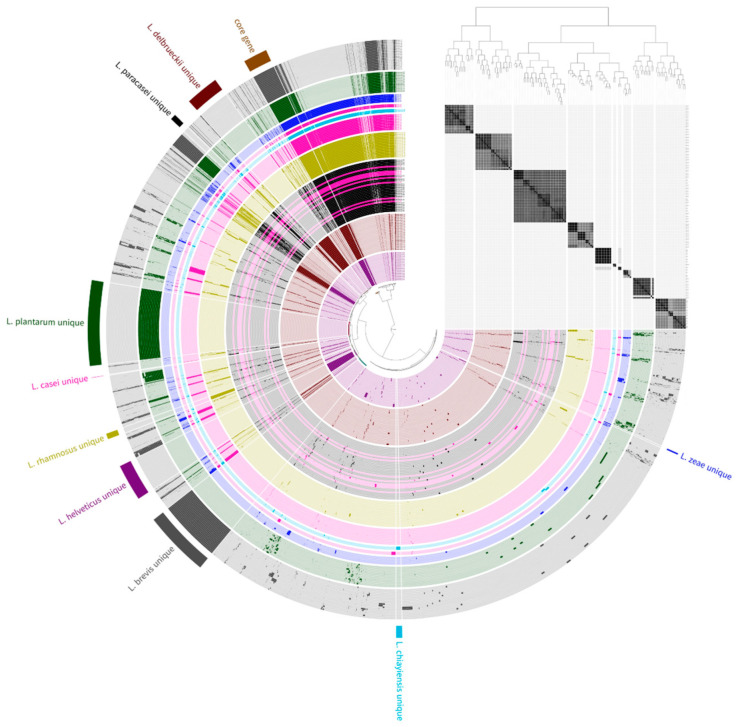
Pan-genome distribution of 141 genome sequences. The blue, sky blue, pink, green, gray, purple, yellow, red, and black color bars represent *L. zeae*, *L. chiayiensis*, *L. casei*, *L. plantarum*, *L. brevis*, *L. helveticus*, *L. rhamnosus*, *L. delbrueckii*, and *L. paracasei* genomes, respectively. The dark and bright of the bar mean gene presence and absence, respectively. The phylogenetic tree constructed based on the gene cluster frequency is on the right.

**Figure 2 foods-10-02112-f002:**
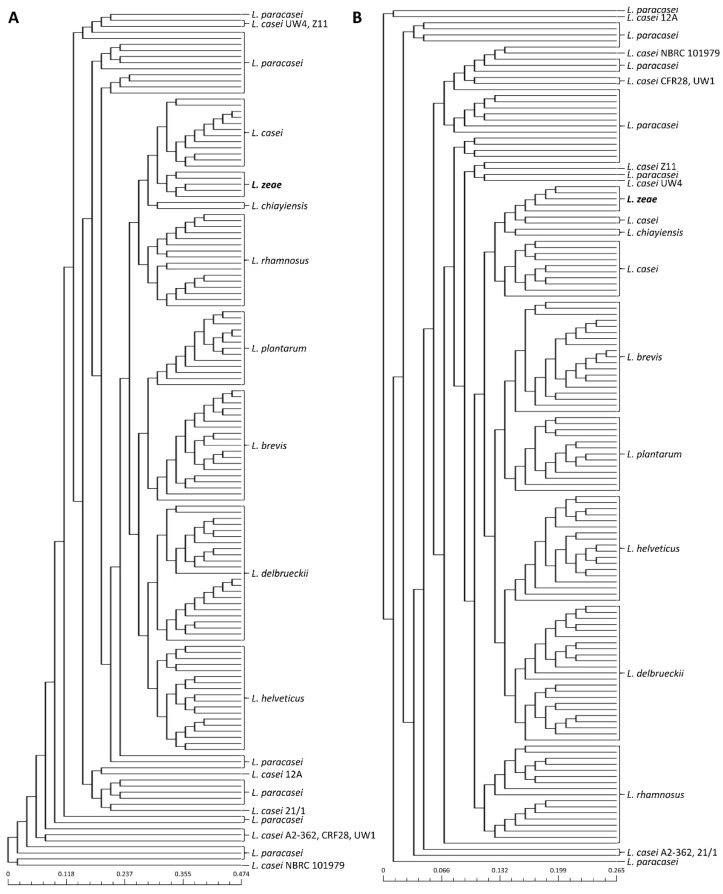
Phylogeny analysis based on (**A**) core-genome and (**B**) pan-genome matrix of 141 genome sequences.

**Figure 3 foods-10-02112-f003:**
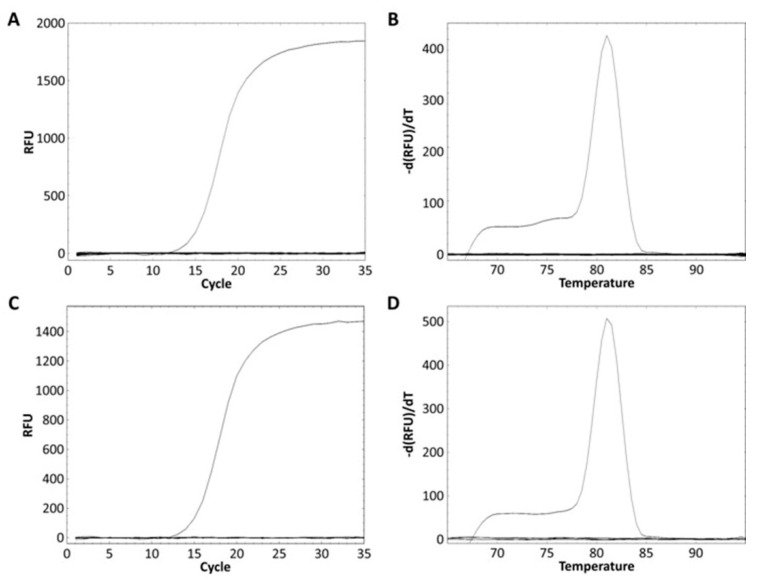
Real-time PCR discrimination of *L. zeae* from the other eight species using the primer. (**A**) Amplification curve and (**B**) melting curve obtained from a pure culture. (**C**) Amplification curve and (**D**) melting curve obtained from the mixed DNA prepared by extracting DNA from the cells of nine species and mixing 25 ng each.

**Figure 4 foods-10-02112-f004:**
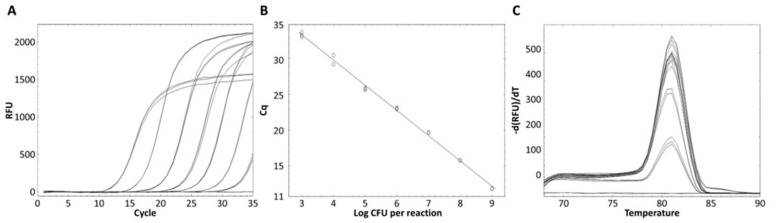
Quantified standard curve of *L. zeae*. (**A**) Amplification plot and (**B**) melting curve obtained by dilutions of *L. zeae* from 10^3^ to 10^9^ CFU/mL. (**C**) Quantified standard curve using the equation y = −3.530x + 36.906 (*R*^2^ = 0.997).

**Figure 5 foods-10-02112-f005:**
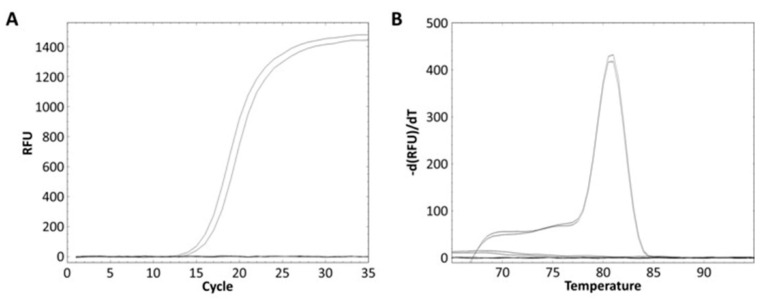
Evaluation of real-time PCR method against 61 bacterial strains. (**A**) All strains except *L. zeae* KACC 11461 and LI220 showed no amplified and (**B**) only these products obtained melting curves.

**Figure 6 foods-10-02112-f006:**
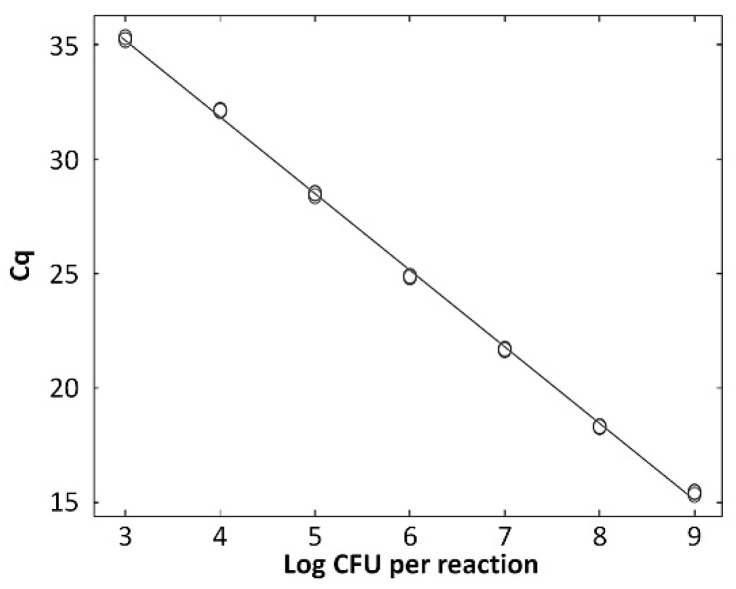
Result for the detection of *L. zeae* in spiked milk sample. Standard curve generated by plotting Cq values with logarithm of the number of *L. zeae* strain artificially inoculated per milliliter of milk. Standard curve equation is y = −3.343 x + 45.215 (*R*^2^ = 0.999).

**Table 1 foods-10-02112-t001:** Genome features used in the analysis.

Species	Strain	Size (Mb)	GC%	CDS	Assembly	Accession No.
*L. zeae*	KCTC 3804	3.11033	47.80	2804	Contig	BACQ01
*L. zeae*	DSM 20178	3.12134	47.70	2812	Scaffold	AZCT01
*L. zeae*	CRBIP24.58	3.09086	47.80	2751	Contig	VBWN01
*L. zeae*	CRBIP24.44	3.08327	47.70	2804	Contig	VBWO01
*L. zeae*	CECT 9104	3.07341	47.97	2753	Complete	LS991421.1
*L. casei* (proposed as *L. paracasei*)	12A	2.90789	46.40	2669	Complete	CP006690.1
*L. casei*	ATCC 393	2.95296	47.86	2606	Complete	AP012544.1
*L. casei*	LC5	3.13287	47.90	2814	Complete	CP017065.1
*L. casei* (proposed as *L. paracasei*)	21/1	3.21588	46.2	3080	Contig	AFYK01
*L. casei* (proposed as *L. paracasei*)	A2-362	3.36127	46.1	3129	Contig	AFYM01
*L. casei* (proposed as *L. paracasei*)	UW4	2.7583	46.4	2519	Contig	AFYS01
*L. casei* (proposed as *L. paracasei*)	Z11	2.74492	46.4	2538	Scaffold	MPOP01
*L. casei*	DS13_13	2.78224	47.7	2453	Contig	QAZE01
*L. casei*	DS1_13	2.84138	47.7	2510	Contig	QAZD01
*L. casei*	YNF-5	2.78037	47.8	2450	Scaffold	SDJZ01
*L. casei*	BCRC 80156	2.82991	47.7	2493	Contig	VBWM01
*L. casei*	BCRC 17487	2.82015	47.7	2489	Contig	VBWL01
*L. casei* (proposed as *L. paracasei*)	NBRC 101979	3.03467	46.1	2826	Contig	BJUH01
*L. casei*	BIO5773	3.08421	47.9	2763	Contig	WBOC01
*L. casei* (proposed as *L. paracasei*)	UW1	3.11557	46.1	2842	Contig	JDWK01
*L. casei* (proposed as *L. paracasei*)	CRF28	3.00996	46.2	2878	Scaffold	JDWL01
*L. casei*	MGB0470	2.94091	47.9	2566	Complete	CP064303.1
*L. casei*	UBLC-42	2.81311	47.7	2496	Contig	JADPYW01
*L. casei*	HUL 5	2.75936	47.8	2482	Contig	JAGDFA01
*L. casei*	HUL 12	2.76045	47.8	2479	Contig	JAGEPP01
*L. chiayiensis*	NCYUAS	2.87209	47.10	2660	Contig	MSSM01
*L. chiayiensis*	BCRC 18859	2.66164	47.30	2146	Contig	NOXN01
*L. paracasei*	TMW 1.1434	3.17011	46.32	2845	Complete	CP016355.1
*L. paracasei*	AO356	3.09656	46.34	2884	Complete	CP025499.1
*L. paracasei*	ATCC 334	2.92433	46.56	2630	Complete	CP000423.1
*L. paracasei*	Zhang	2.89846	46.42	2625	Complete	CP001084.2
*L. paracasei*	BL23	3.0792	46.30	2885	Complete	FM177140.1
*L. paracasei*	8700:02:00	3.02535	46.30	2784	Complete	CP002391.1
*L. paracasei*	BD-II	3.12729	46.25	2919	Complete	CP002618.1
*L. paracasei*	LC2W	3.07743	46.35	2859	Complete	CP002616.1
*L. paracasei*	W56	3.1321	46.25	2843	Complete	HE970764.1
*L. paracasei*	LOCK919	3.14337	46.18	2928	Complete	CP005486.1
*L. paracasei*	N1115	3.06428	46.46	2809	Complete	CP007122.1
*L. paracasei*	JCM 8130	3.0178	46.56	2770	Complete	AP012541.1
*L. paracasei*	CAUH35	2.97335	46.33	2712	Complete	CP012187.1
*L. paracasei*	L9	3.07644	46.30	2791	Complete	CP012148.1
*L. paracasei*	KL1	2.91889	46.60	2702	Complete	CP013921.1
*L. paracasei*	IIA	3.24614	46.22	3049	Complete	CP014985.1
*L. paracasei*	TK1501	2.94254	46.50	2720	Complete	CP017716.1
*L. paracasei*	FAM18149	2.96971	46.34	2768	Complete	CP017261.1
*L. paracasei*	EG9	3.07441	46.44	2789	Complete	CP029546.1
*L. paracasei*	Lpc10	3.05212	46.30	2780	Complete	CP029686.1
*L. paracasei*	ZFM54	3.04868	46.35	2820	Complete	CP032637.1
*L. paracasei*	IJH-SONE68	3.1812	46.42	2847	Complete	AP018392.1
*L. paracasei*	SRCM103299	3.18745	46.41	2924	Complete	CP035563.1
*L. paracasei*	CBA3611	3.10253	46.34	2890	Complete	CP041657.1
*L. paracasei*	NJ	3.08341	46.40	2763	Complete	CP041944.1
*L. rhamnosus*	ATCC 53103	3.00505	46.70	2685	Complete	AP011548.1
*L. rhamnosus*	GG (ATCC 53103)	3.01011	46.70	2706	Complete	FM179322.1
*L. rhamnosus*	Lc 705	3.03311	46.63	2652	Complete	FM179323.1
*L. rhamnosus*	ATCC 8530	2.96034	46.80	2693	Complete	CP003094.1
*L. rhamnosus*	LOCK900	2.88338	46.80	2583	Complete	CP005484.1
*L. rhamnosus*	LOCK908	2.9909	46.80	2720	Complete	CP005485.1
*L. rhamnosus*	LRB	2.93495	46.80	2428	Complete	CP016823.1
*L. rhamnosus*	BFE5264	3.11476	46.76	2862	Complete	CP014201.1
*L. rhamnosus*	Pen	2.88497	46.80	2638	Complete	CP020464.1
*L. rhamnosus*	4B15	3.04784	46.70	2729	Complete	CP021426.1
*L. rhamnosus*	LR5	2.97259	46.70	2710	Complete	CP017063.1
*L. rhamnosus*	DSM 14870	3.01315	46.70	2761	Complete	CP006804.1
*L. rhamnosus*	GG	3.01012	46.70	2770	Complete	CP031290.1
*L. rhamnosus*	LR-B1	3.0075	46.70	2800	Complete	CP025428.1
*L. rhamnosus*	NCTC13710	2.99105	46.80	2764	Complete	LR134322.1
*L. rhamnosus*	NCTC13764	2.98839	46.80	2765	Complete	LR134331.1
*Levilactobacillus brevis*	ATCC 367	2.34023	46.04	2178	Complete	CP000416.1
*L. brevis*	KB290	2.58788	45.57	2449	Complete	AP012167.1
*L. brevis*	NPS-QW-145	2.55267	45.80	2386	Complete	CP015398.1
*L. brevis*	TMW 1.2112	2.67355	45.72	2331	Complete	CP016797.1
*L. brevis*	TMW 1.2113	2.66787	45.70	2326	Complete	CP019750.1
*L. brevis*	TMW 1.2108	2.91798	45.27	2746	Complete	CP019734.1
*L. brevis*	TMW 1.2111	2.88201	45.31	2513	Complete	CP019743.1
*L. brevis*	100D8	2.47773	45.75	2337	Complete	CP015338.1
*L. brevis*	SRCM101174	2.57125	45.59	2427	Complete	CP021479.1
*L. brevis*	SRCM101106	2.55412	45.60	2396	Complete	CP021674.1
*L. brevis*	BDGP6	2.78511	45.60	2671	Complete	CP024635.1
*L. brevis*	ZLB004	2.65509	45.61	2409	Complete	CP021456.1
*L. brevis*	UCCLBBS124	2.72824	45.62	2575	Complete	CP031169.1
*L. brevis*	SA-C12	2.50886	45.72	2337	Complete	CP031185.1
*L. brevis*	UCCLB556	2.56198	45.74	2347	Complete	CP031174.1
*L. brevis*	UCCLB95	2.52877	45.88	2233	Complete	CP031182.1
*L. brevis*	UCCLBBS449	2.77507	45.45	2571	Complete	CP031198.1
*L. brevis*	UCCLB521	2.41605	45.92	2186	Complete	CP031208.1
*L. brevis*	NCTC13768	2.49433	46.00	2358	Complete	LS483405.1
*Lactobacillus delbrueckii*	ATCC BAA-365	1.85695	49.70	1593	Complete	CP000412.1
*L. delbrueckii*	ATCC 11842	1.865	49.70	1568	Complete	CR954253.1
*L. delbrueckii*	ND02	2.13198	49.59	1841	Complete	CP002341.1
*L. delbrueckii*	2038	1.87292	49.70	1562	Complete	CP000156.1
*L. delbrueckii*	MN-BM-F01	1.87507	49.70	1585	Complete	CP013610.1
*L. delbrueckii*	KCCM 34717	2.26338	49.10	1891	Complete	CP018215.1
*L. delbrueckii*	DSM 26046	1.8918	50.10	1614	Complete	CP018218.1
*L. delbrueckii*	KCTC 13731	1.91051	50.00	1600	Complete	CP018216.1
*L. delbrueckii*	JCM 17838	2.00434	50.10	1726	Complete	CP018217.1
*L. delbrueckii*	KCTC 3035	1.97273	50.00	1697	Complete	CP018156.1
*L. delbrueckii*	JCM 15610	2.02186	49.37	1694	Complete	CP018614.1
*L. delbrueckii*	DSM 20080	1.86818	49.80	1564	Complete	CP019120.1
*L. delbrueckii*	ND04	1.86175	49.60	1538	Complete	CP016393.1
*L. delbrueckii*	TUA4408L	2.01244	49.90	1718	Complete	CP021136.1
*L. delbrueckii*	DSM 20072	2.16598	49.00	1800	Complete	CP022988.1
*L. delbrueckii*	KCTC 3034	2.23761	49.00	1889	Complete	CP023139.1
*L. delbrueckii*	L99	1.84811	49.70	1575	Complete	CP017235.1
*L. delbrueckii*	KLDS1.0207	1.86918	49.80	1620	Complete	CP032451.1
*L. delbrueckii*	NWC_1_2	2.25977	48.58	1909	Complete	CP029250.1
*L. delbrueckii*	KLDS1.1011	1.88749	49.80	1629	Complete	CP041280.1
*L. delbrueckii*	NBRC 3202	1.91031	50.10	1636	Complete	AP019750.1
*L. delbrueckii*	ACA-DC 87	1.856	49.80	1582	Complete	LT899687.1
*L. delbrueckii*	lactis1	2.05032	49.60	1675	Complete	LS991409.1
*Lactobacillus helveticus*	DPC 4571	2.08093	37.10	1700	Complete	CP000517.1
*L. helveticus*	R0052	2.12921	36.80	1743	Complete	CP003799.1
*L. helveticus*	H10	2.17238	36.80	1863	Complete	CP002429.1
*L. helveticus*	CNRZ32	2.22596	36.90	1854	Complete	CP002081.1
*L. helveticus*	H9	1.87112	37.00	1531	Complete	CP002427.1
*L. helveticus*	KLDS1.8701	2.10663	36.89	1723	Complete	CP009907.1
*L. helveticus*	MB2-1	2.08406	36.90	1755	Complete	CP011386.1
*L. helveticus*	CAUH18	2.16058	36.80	1840	Complete	CP012381.1
*L. helveticus*	D76	2.05832	37.00	1660	Complete	CP016827.1
*L. helveticus*	D75	2.05307	37.00	1659	Complete	CP020029.1
*L. helveticus*	FAM8627	2.04903	36.99	1666	Complete	CP015444.1
*L. helveticus*	FAM8105	2.25524	37.04	1881	Complete	CP015496.1
*L. helveticus*	FAM22155	2.19866	37.09	1817	Complete	CP015498.1
*L. helveticus*	LH99	2.08274	37.10	1749	Complete	CP017982.1
*L. helveticus*	NWC_2_3	2.23298	37.54	1812	Complete	CP031016.1
*L. helveticus*	NWC_2_4	2.23013	37.40	1782	Complete	CP031018.1
*L. helveticus*	LH5	2.16368	36.81	1858	Complete	CP019581.1
*L. helveticus*	IDCC3801	2.15725	36.82	1853	Complete	CP035307.1
*Lactiplantibacillus plantarum*	ST-III	3.30794	44.50	2967	Complete	CP002222.1
*L. plantarum*	LB1-2	3.54187	44.11	3227	Complete	CP025991.1
*L. plantarum*	DSM 16365	3.35034	44.94	2999	Complete	CP032751.1
*L. plantarum*	WCFS1	3.34862	44.45	3014	Complete	AL935263.2
*L. plantarum*	ATCC 8014	3.30947	44.43	2972	Complete	CP024413.1
*L. plantarum*	ATG-K6	3.2625	44.50	2926	Complete	CP032464.1
*L. plantarum*	ZFM9	3.43401	44.28	3084	Complete	CP032642.1
*L. plantarum*	NCIMB700965.EF.A	3.21713	44.54	2762	Complete	CP026505.1
*L. plantarum*	UNQLp11	3.53493	44.20	3164	Complete	CP031140.1
*L. plantarum*	TMW 1.1308	3.33353	44.51	2937	Complete	CP021929.1
*L. plantarum*	KCCP11226	3.3821	44.39	3046	Complete	CP046262.1
*L. plantarum*	8P-A3	3.33278	44.38	2982	Complete	CP046726.1
*L. plantarum*	SRCM101511	3.27272	44.41	2907	Complete	CP028235.1

**Table 2 foods-10-02112-t002:** List of bacterial strains used in this study.

Species	Strain Number
*L. zeae*	KACC ^1^ 11461 ^T^
*L. zeae*	LI ^2^ 220
*L. casei*	KCTC ^3^ 3109 ^T^
*L. casei*	KCTC 13086
*L. casei*	KCTC 3110 ^T^
*L. chiayiensis*	NBRC ^4^ 112906 ^T^
*L. paracasei*	MGB ^5^ 0543
*L. paracasei*	KCTC 3165 ^T^
*L. paracasei*	KACC 12427 ^T^
*L. rhamnosus*	KCTC 5033 ^T^
*L. rhamnosus*	KCTC 3237 ^T^
*L. rhamnosus*	KCTC 13088
*Amylolactobacillus amylophilus*	KACC 11430 ^T^
*Apilactobacillus kunkeei*	KACC 19371 ^T^
*Companilactobacillus crustorum*	KACC 16344 ^T^
*Companilactobacillus farciminis*	KACC 12423 ^T^
*Companilactobacillus heilongjiangensis*	KACC 18741 ^T^
*Fructilactobacillus lindneri*	KACC 12445 ^T^
*Fructilactobacillus sanfranciscensis*	KACC 12431 ^T^
*Lactiplantibacillus paraplantarum*	KACC 12373 ^T^
*Lactiplantibacillus pentosus*	KACC 12428 ^T^
*L. plantarum*	KACC 11451 ^T^
*L. plantarum*	KACC 12404 ^T^
*L. plantarum*	KACC 15357
*Lactobacillus acetotolerans*	KACC 12447 ^T^
*Lactobacillus acidophilus*	KACC 12419 ^T^
*Lactobacillus amylolyticus*	KACC 12374 ^T^
*Lactobacillus amylovorus*	KACC 12435 ^T^
*L. delbrueckii* subsp. *bulgaricus*	KACC 12420 ^T^
*L. delbrueckii* subsp. *delbrueckii*	KACC 13439 ^T^
*L. delbrueckii* subsp. *lactis*	KACC 12417 ^T^
*Lactobacillus gallinarum*	KACC 12370 ^T^
*Lactobacillus gasseri*	KCTC 3163 ^T^
*L. helveticus*	KACC 12418 ^T^
*Lactobacillus jensenii*	KCTC 5194 ^T^
*Lactobacillus johnsonii*	KCTC 3801 ^T^
*Latilactobacillus curvatus*	KACC 12415 ^T^
*Latilactobacillus sakei*	KCTC 3603 ^T^
*Lentilactobacillus buchneri*	KACC 12416 ^T^
*Lentilactobacillus parabuchneri*	KACC 12363 ^T^
*L. brevis*	KCTC 3498 ^T^
*Levilactobacillus zymae*	KACC 16349 ^T^
*Ligilactobacillus acidipiscis*	KACC 12394 ^T^
*Ligilactobacillus agilis*	KACC 12433 ^T^
*Ligilactobacillus ruminis*	KACC 12429 ^T^
*Ligilactobacillus salivarius*	KCTC 3600
*Limosilactobacillus fermentum*	KACC 11441 ^T^
*Limosilactobacillus mucosae*	KACC 12381 ^T^
*Limosilactobacillus reuteri*	KCTC 3594 ^T^
*Loigolactobacillus coryniformis*	KACC 12411 ^T^
*Lactococcus lactis*	KCTC 3769 ^T^
*Bifidobacterium animalis* subsp.* animalis*	KACC 16637 ^T^
*Bifidobacterium animalis* subsp. *lactis*	KACC 16638 ^T^
*Bifidobacterium bifidum*	KCTC 3418
*Bifidobacterium bifidum*	KCTC 3440
*Bifidobacterium breve*	KACC 16639 ^T^
*Bifidobacterium breve*	KCTC 3419
*Bifidobacterium longum* subsp. *infantis*	KCTC 3249 ^T^
*Bifidobacterium longum* subsp. *longum*	KCCM ^6^ 11953 ^T^
*Enterococcus faecalis*	KCTC 3206 ^T^
*Enterococcus faecium*	KCTC 13225 ^T^

^T^, type strain; ^1^ KACC, the Korean Agricultural Culture Collection; ^2^ LI, the Laboratory Isolate; ^3^ KCTC, the Korean Collection for Type Cultures; ^4^ NBRC, the NITE Biological Resource Center; ^5^ MGB, the Microorganism and Gene Bank; ^6^ KCCM, the Korean Culture Center of Microorganisms.

**Table 3 foods-10-02112-t003:** Primer information for *L. zeae* designed in this study.

Species	Primer Name	Sequence (5’–3’)	Product Size (bp)
*L. zeae*	Zeae-F	CAT GGC CGA TAT GCA GCA TT	128
	Zeae-R	GAT CTG CCA GGT TCC ATG AC	

**Table 4 foods-10-02112-t004:** Comparison of 16S rRNA gene sequencing and real-time PCR.

Strains	16S rRNA Gene Sequencing ^1^	Real-Time PCR ^2^
KACC 11461	*L. zeae* (LS991421.1, 99.8%)	*L. zeae* (Ct 16.31)
	*L. rhamnosus* (MG984549.1, 99.8%)	
	*L. casei* (CP017065.1, 99.8%)	
LI 220	*L. zeae* (LS991421.1, 99.9%)	*L. zeae* (Ct 17.09)
	*L. rhamnosus* (MG984549.1, 99.9%)	
	*L. casei* (AP012544.1, 99.9%)	

^1^ Description identified by 16S rRNA gene sequencing (accession no., % identity); ^2^ Detected species by real-time PCR developed in this study (Ct value).

## Data Availability

The data presented in this study are available on request from the corresponding author.

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
