# Peer review of "Differentiation of Lacticaseibacillus zeae Using Pan-Genome Analysis and Real-Time PCR Method Targeting a Unique Gene"

_foods, 2021, doi:10.3390/foods10092112_

Round 1

Reviewer 1 Report

This manuscript is one in a series from the Kim laboratory which uses genome mapping to detect genes which are unique to specific species of lactic acid bacteria (LAB)( see references 12, 13 and 21).  The utility in this approach is seen in the application of RT-PCR methods to differentiate between species in fermented foods which contain complex and dynamic communities.   The focus of the current study is detecting and quantifying L. zeae, a species with a confused history in bacterial nomenclature and recently proposed as a separate species within the ‘casei’ group of Lacticaseibacillus.  The strategy to identify biomarker genes for specific species involved comparing the pan-genomes of 124 sequenced genomes, given that core genomes encode functions likely to be found across species.  This is a reasonable approach and relies heavily on:

  • the accuracy of the initial species classification of the strains examined and
  • the assumption that all examples of the species contain the gene and that the gene is absent in all other related and non-related bacteria.

Herein lies some of the difficulties in the current submission, as detailed below.

The main points to be addressed are:

  1. The authors used 124 published genome assemblies from NCBI (Table 1). However, 51 of these were casei group members (5 zeae, 3 casei, 2 chiayiensis, 25 paracasei and 16 rhamnosus).  The remainder were included to demonstrate that the proposed marker, KRK10099.1, does not occur in other species of LAB.  However, zeae is most closely related to L. casei (Huang et al. references) and the L. casei type strain (ATCC393) is also related to L. rhamnosus (based on 16S-23S intergenic regions, personal observation and Kwon et al., 2004).  However, one of the L. casei (12A) in Table 1 is actually L. paracasei (genome BLAST/NCBI and ANI paracasei clique, IMG/G).  This is clear from Fig. 2, which shows one ‘L. casei’ clusters with paracasei – which should have alerted the authors to the fault in nomenclature.

The authors are aware that the taxonomic position of many strains in the NCBI database is not correct, likely due to many strains that were named prior to the availability of a large number of genome assemblies – and NCBI has not updated many L. casei that are actually L. paracasei.  This can be checked by examining the assembly details.  However, having only two strains of valid L. casei for the basis of comparing genomes is simply not sufficient, given the confused phylogeny of this group and incorrect nomenclature.  This point must be addressed by the authors e.g. by including a broader selection of L. casei which have been correctly named as such.  Please note that  ‘L. casei’ assembly GCA_001940585 is actually for a strain of L. zeae, adding to the confusion when selecting representative genome assemblies.

  1. The stains used to identify genes unique to species (notwithstanding point 1) came from Table 1 and validation of the PCR method employed strains from Table 2 (obtained from local culture collections) and from one suspected zeae from a milk sample.

The authors do not indicate how the species listed in Table 2 were classified (genome sequencing or simply named when deposited/collected and the names assumed to be correct?)  What methods were used to confirm nomenclature?  The authors correctly point out (lines 47-48) that 16S rRNA gene sequencing is not sufficient to differentiate between species in the casei group, yet this is the method used to validate the species listed in Table 2 (section 2.4) and to identify one suspected L. zeae from milk (lines 94-95).  What differentiating traits made the author suspect that strain LI220 was L. zeae?  This is important, as this strain was used to demonstrate the accuracy of the PCR method in identifying L. zeae (Table 4) together with one strain from Table 2.  The authors correctly point out that isolates of L. zeae are rare in fermented foods/milk but the example of one or two isolates validating the utility of the method is a little thin, notwithstanding validating the method using some of the strains in Table 2 (see below regarding how this was done). 

  1. The proposed marker, KRK10099.1, is claimed to be unique to zeae. This is probably a valid claim.  However, BLAST of the marker’s protein sequence shows similarity with several LAB species, including L. casei group strains and enterococci, leuconostoc and lactococci. The level of similarity is low (about 30% for the top 11 hits in KEGG) and there is conservation of the domain that specifies the glycosyl transferase.  This is common in the firmicutes where modular proteins share a common evolutionary track but considerable sequence diversity. The similarity cut-off point was set at the default of 50%, which is reasonable.  However, there is little comment on this (lines 171-173):  some further detail is required that explains the relationship between the genes in the different species AND the fit of the PCR primer design within the gene must be clarified (so it is clear that the amplified fragment would be unlikely to be found in the closest bacterial relatives.  Species which contain gene homologs were not tested to prove specificity of the PCR method to L. zeae.  This is needed.

Some clustal omega analysis demonstrating similarity (or not) between the glycosyl transferases in other LAB species could be included in supplementary data, given there is no data to substantiate the comments in lines 171-173.

  1. The methods are not totally clear, as the authors are mixing up ‘culture’ with ‘cells’, I believe, and consequently what methods were used in some of the Tables/Figures is not clear. For example, the method for demonstrating that the primers used to amplify the glycosyl transferase gene  is said to have used 20 ng of DNA template from each strain being examined (section 2.3) – did the authors prepare ‘dummy’ mixtures of cells, extract the DNA then test what was amplified?  This would better represent how PCR would be applied in food samples.   Please provide more detail in section 2.3 and clarify the methods used for later Tables and Figures.

The authors have not validated the PCR method by seeding milk, for example, with test and control strains – further validation is mentioned but precisely what is to come is not stated.

An annotated pdf is provided to point out precise sentences or areas where some clarity is sought.  

Overall:

The manuscript has merit but the technical matters described above require addressing, as the genome analysis requires further comparisons with L. casei strains, including those where a homolog comes up in KEGG BLASTp, and demonstrating that the primers developed in this study do not detect these homologs in species not in the mix already tested.

Author Response

Response to Reviewer 1 Comments

This manuscript is one in a series from the Kim laboratory which uses genome mapping to detect genes which are unique to specific species of lactic acid bacteria (LAB) (see references 12, 13 and 21). The utility in this approach is seen in the application of RT-PCR methods to differentiate between species in fermented foods which contain complex and dynamic communities. The focus of the current study is detecting and quantifying L. zeae, a species with a confused history in bacterial nomenclature and recently proposed as a separate species within the ‘casei’ group of Lacticaseibacillus. The strategy to identify biomarker genes for specific species involved comparing the pan-genomes of 124 sequenced genomes, given that core genomes encode functions likely to be found across species. This is a reasonable approach and relies heavily on:

  • the accuracy of the initial species classification of the strains examined and
  • the assumption that all examples of the species contain the gene and that the gene is absent in all other related and non-related bacteria.

Herein lies some of the difficulties in the current submission, as detailed below.

The main points to be addressed are:

  1. The authors used 124 published genome assemblies from NCBI (Table 1). However, 51 of these were casei group members (5 zeae, 3 casei, 2 chiayiensis, 25 paracasei and 16 rhamnosus). The remainder were included to demonstrate that the proposed marker, KRK10099.1, does not occur in other species of LAB. However, zeaeis most closely related to L. casei(Huang et al. references) and the L. casei type strain (ATCC393) is also related to L. rhamnosus (based on 16S-23S intergenic regions, personal observation and Kwon et al., 2004). However, one of the L. casei (12A) in Table 1 is actually L. paracasei (genome BLAST/NCBI and ANI paracasei clique, IMG/G). This is clear from Fig. 2, which shows one ‘L. casei’ clusters with paracasei – which should have alerted the authors to the fault in nomenclature.

Response: Thank you for your comment. As you recommended, we marked the genomes that needed reclassification in Table 1 and added the sentence in lines 148-151 as follows:

Lines 148-151: However, eight genomes of L. casei were clustered with L. paracasei instead of L. casei. Consistent with the results of a previous study [25], we suggested that L. casei 12A, 12/1, A2-362, UW4, Z11, NBRC 101979, UW1, and CRF28 should be moved out and placed into the L. paracasei.

Table 1: We marked the genomes that needed reclassification in Table 1.

The authors are aware that the taxonomic position of many strains in the NCBI database is not correct, likely due to many strains that were named prior to the availability of a large number of genome assemblies – and NCBI has not updated many L. casei that are actually L. paracasei. This can be checked by examining the assembly details. However, having only two strains of valid L. casei for the basis of comparing genomes is simply not sufficient, given the confused phylogeny of this group and incorrect nomenclature. This point must be addressed by the authors e.g. by including a broader selection of L. casei which have been correctly named as such. Please note that ‘L. casei’ assembly GCA_001940585 is actually for a strain of L. zeae, adding to the confusion when selecting representative genome assemblies.

Response: As you recommended, we re-analyzed the pangenome by adding the 17 genomes of L. casei (total 20 genomes) and demonstrated specificity. The added genome information and reanalysis results were added to the in lines 13, 15, 65, 67, 74, 159, 165, 167, 167-168, 170-171, and 174 as follows:

Line 13, 67, 167: 141 genomes

Line 65, 74, 159, 165: 141 genome sequences

Line 13: yielded an 17,978 pan-genome

Line 15: 136 other genomes

Lines 167-168: The 17,978 pan-genome obtained from 141 genomes is composed of 144 core genes and 4,271 unique genes

Lines 170-171: biogenesis (38.6%), and nucleotide transport & metabolism (6.7%).

Line 174: Among the 13,563 accessory genes

Table 1: We added genome features of the newly 17 L. casei in Table 1.

Figure 1: We added pan-genome distribution for L. casei

Figure 2: We added phylogeny analysis based on core-genome and pan-genome for L. casei

  1. The stains used to identify genes unique to species (notwithstanding point 1) came from Table 1 and validation of the PCR method employed strains from Table 2 (obtained from local culture collections) and from one suspected zeaefrom a milk sample.
    The authors do not indicate how the species listed in Table 2 were classified (genome sequencing or simply named when deposited/collected and the names assumed to be correct?) What methods were used to confirm nomenclature? The authors correctly point out (lines 47-48) that 16S rRNA gene sequencing is not sufficient to differentiate between species in the caseigroup, yet this is the method used to validate the species listed in Table 2 (section 2.4) and to identify one suspected L. zeae from milk (lines 94-95). What differentiating traits made the author suspect that strain LI220 was L. zeae? This is important, as this strain was used to demonstrate the accuracy of the PCR method in identifying L. zeae (Table 4) together with one strain from Table 2. The authors correctly point out that isolates of L. zeae are rare in fermented foods/milk but the example of one or two isolates validating the utility of the method is a little thin, notwithstanding validating the method using some of the strains in Table 2 (see below regarding how this was done).

Response: By definition, type strain is descendants of the original isolates used in species and subspecies descriptions, as defined by the Bacteriological Code, that exhibit all of the relevant phenotypic and genotypic properties cited in the original published taxonomic circumscriptions (Kyrpides et al., doi: 10.1371/journal.pbio.1001920). In general, a strain designation including superscript T shows the strain is a type strain. Most of the strains listed in Table 2 are type strains, so we are confident that the named species in Table 2 are accurately classified. Among the 12 Lacticaseibacillus strains, L. zeae LI220 is a laboratory isolate, and L. casei KCTC 13086, L. paracasei MGB 0543, and L. rhamnosus KCTC 13088 is not type strains, but has been applied to several studies. Culture collection confirmed these species by 16S rRNA gene sequencing method or whole-genome sequencing. Among the strains listed in Table 2, we denoted the type strain as T.

Line 98: T, type strain

Table 2: We indicated the type strain.

  1. The proposed marker, KRK10099.1, is claimed to be unique to zeae. This is probably a valid claim. However, BLAST of the marker’s protein sequence shows similarity with several LAB species, including L. caseigroup strains and enterococci, leuconostoc and lactococci. The level of similarity is low (about 30% for the top 11 hits in KEGG) and there is conservation of the domain that specifies the glycosyl transferase. This is common in the firmicutes where modular proteins share a common evolutionary track but considerable sequence diversity. The similarity cut-off point was set at the default of 50%, which is reasonable. However, there is little comment on this (lines 171-173): some further detail is required that explains the relationship between the genes in the different species AND the fit of the PCR primer design within the gene must be clarified (so it is clear that the amplified fragment would be unlikely to be found in the closest bacterial relatives. Species which contain gene homologs were not tested to prove specificity of the PCR method to L. zeae. This is needed.
    Some clustal omega analysis demonstrating similarity (or not) between the glycosyl transferases in other LAB species could be included in supplementary data, given there is no data to substantiate the comments in lines 171-173.

Response: Thank you for your comment. The specific gene of L. zeae showed within 36% homologies with other strains, such as Enterococcus, Lacticaseibacillus, and Lactococcus. However, the protein sequences did not match continuously with other strains. Also, the nucleotide sequence of glycosyltransferase shows less than 3.4% homologies with other strains. Therefore, the specificity did not change even when the primer design outside the regions of sequence similarity. As you recommended, we newly performed an analysis demonstrating similarity between the glycosyltransferase in L. zeae and other strains and added the sentence and supplementary material in lines 86-89, 181-184, and 199-201 as follows:

Lines 86-89: The unique gene for L. zeae was compared with other strains through BLASTP search against the Kyoto Encyclopedia of Genes and Genomes (KEGG) database, and sequence alignment was performed using Clustal Omega (https://www.ebi.ac.uk/Tools/msa/clustalo/).

Lines 181-184: The glycosyltransferase of L. zeae was compared with other bacterial strains, and as a result, it showed the highest homology with Enterococcus gallinarum (36% identity) and less than 34% homology with other Lacticaseibacillus strains (Table S1).

Lines 199-201: As shown in Figure S1, since the sequences did not match consecutively, designing a primer at any position within this gene does not result in amplification in other bacteria.

Table S1: We newly added Table S1 for similarity for glycosyltransferase between L. zeae and other strains.

Figure S1: We newly added Figure S1 for sequence alignment for glycosyltransferase between L. zeae and other strains.

  1. The methods are not totally clear, as the authors are mixing up ‘culture’ with ‘cells’, I believe, and consequently what methods were used in some of the Tables/Figures is not clear. For example, the method for demonstrating that the primers used to amplify the glycosyl transferase gene is said to have used 20 ng of DNA template from each strain being examined (section 2.3) – did the authors prepare ‘dummy’ mixtures of cells, extract the DNA then test what was amplified? This would better represent how PCR would be applied in food samples. Please provide more detail in section 2.3 and clarify the methods used for later Tables and Figures.

Response: Thank you for your comment. In this study, cultures were not used, but 20 ng of DNA extracted from cells of nine species was mixed. As you recommended, we revised the sentence in lines 16, 117-120, and 219 as follows:

Line 16 and 219: mixed DNA

Lines 117-120: pure and mixed DNA of nine species, including species mainly found in raw milk and closely related to L. zeae. For the preparation of mixed DNA, 20 ng of DNA was extracted from cells of nine species and mixed.

The authors have not validated the PCR method by seeding milk, for example, with test and control strains – further validation is mentioned but precisely what is to come is not stated.

Response: Thank you for your comment. As you recommended, we newly tested food sample artificially inoculated with L. zeae and added the sentence for the test results in lines 127, 134-139, and 266-273 as follows:

Line 127: 2.4. Evaluation and application of real-time PCR

Lines 134-139: Our method was applied to spiked food sample. The milk sample was purchased from a local market in Korea and was confirmed to be free of L. zeae. A spiked milk sample was prepared according to a previous study [21,22]. Briefly, 25 ml milk sample was inoculated with L. zeae at a concentration of 103 to 109 CFU/ml. Genomic DNA was extracted inoculated sample according to the method described above, and real-time PCR was performed.

Lines 266-273: The quantification of genomic DNA in a food sample was conducted by artificially adding L. zeae strain to milk. The real-time PCR method developed in this study could successfully be used to identify L. zeae at a concentration of 103 to 109 CFU/ml in milk (Figure 6). Samples artificially inoculated with L. zeae (Ct values: 14.97 to 34.78) had a slightly higher Ct value than the pure culture of L. zeae strain, which seemed to slightly affect the efficiency of real-time PCR. This may be due to the presence of PCR inhibitors from fast and protein in food [21]. Therefore, our real-time PCR method was able to identify L. zeae in the food matrix.

Figure 6: We newly added the result for the detection of L. zeae in spiked milk sample in Figure 6.

Overall:

The manuscript has merit but the technical matters described above require addressing, as the genome analysis requires further comparisons with L. casei strains, including those where a homolog comes up in KEGG BLASTp, and demonstrating that the primers developed in this study do not detect these homologs in species not in the mix already tested.

Response: Thank you for your comment. As you recommended, we newly tested food sample artificially inoculated with L. zeae, and newly tested the presence of other LAB species which may contain similar genes.

Response to Reviewer 1 Comments (PDF comments): Please also see the attachment

  1. Line 54: Huang et al. DOI 10.1099/ijsem.0.003969 used multi-locus sequence analysis so this statement is not totally accurate. The sentence is also confusing as it goes on to say 'using whole genome sequences', but MLST does not need whole genome analysis and the manuscript argues that detecting the unique marker by RT-PCR doesn't need whole genome sequences.

Response: Thank you for your critical comments. As you recommended, we revised the sentence in lines 57-59 as follows:

Lines 57-59: However, the development of the PCR method to distinguish L. zeae from other closely related species using a marker obtained based on the pangenome has rarely been reported.

  1. Line 61: NCBI has not update the names of lactobacilli when published evidence or Genome BLAST would place the names into a different species. Consequently, the list in Table 1 includes strain 12A as 'L. casei' whereas NCBI BLAST would place this as L. paracasei - also see paper by Ghosh et al. doi:10.3390/microorganisms7110487, which proposed reclassification of many L. casei as L. paracasei.

Response: As you recommended, we marked the genomes that needed reclassification in Table 1 and added the sentence in lines 148-151 and modified Table 1:

Lines 148-151: However, eight genomes of L. casei were clustered with L. paracasei instead of L. casei. Consistent with the results of a previous study [25], we suggested that L. casei 12A, 12/1, A2-362, UW4, Z11, NBRC 101979, UW1, and CRF28 should be moved out and placed into the L. paracasei.

  1. Line 72-74: Please review this sentence - ?version?. Proteins are classified on the basis of their COGs so that like functions are then grouped. Please explain precisely what you did, given that many proteins can fit more than one COG classification and many proteins in the lactobacilli genomes have no COG assigned, which means you restricted the analysis to known protein types.

Response: As you recommended, we revised the sentence and added the description for the COG analysis in lines 79-83 as follows:

Lines 79-83: cluster of orthologous groups (COG) database (http://www.ncbi.nlm.nih.gov/COG/) to assign categories to representative protein sequences. Since some proteins in Lactobacilli genomes can fit more than one COG classification, and some proteins have no COG assigned, COG analysis restricted the analysis to known protein types.

  1. Line 76: Please provide an explanation of what you mean, exactly, when referring to the 'accessory-genome'. Non-core? The pan-and core-genome parts make up one genome, and an 'accessory' may be considered to be outside of this e.g. a plasmid.

Response: As you recommended, we added the explanation of accessory-genome in line 84 as follows:

Line 84: accessory-genome, the set of proteins present in some, but not all genomes,

  1. Table 1: Genome BLAST/NCBI places this strain, 12A, with other strains that are 'paracasei' based on many bioinformatic tools (see ANI clusters in the IMG/M database).

Response: As you recommended, we marked the genomes that needed reclassification in Table 1 and added the sentence in lines 148-151 as follows:

Lines 148-151: However, eight genomes of L. casei were clustered with L. paracasei instead of L. casei. Consistent with the results of a previous study [25], we suggested that L. casei 12A, 12/1, A2-362, UW4, Z11, NBRC 101979, UW1, and CRF28 should be moved out and placed into the L. paracasei.

  1. Table 2: It is very difficult to relate this list to Table 1. Table 1 is very useful, as it quotes the NCBI assembly numbers. However, there are no assembly numbers for the strains in this list AND given the later statement (lines 125-126) that the genomes are misnamed, how confident are the authors that the named species in Table 2 are, indeed, accurately classified/named? Or were names accepted as accurate? how did the culture collections confirm the species?

Response: By definition, type strain is descendants of the original isolates used in species and subspecies descriptions, as defined by the Bacteriological Code, that exhibit all of the relevant phenotypic and genotypic properties cited in the original published taxonomic circumscriptions (Kyrpides et al., doi: 10.1371/journal.pbio.1001920). In general, a strain designation including superscript T shows the strain is a type strain. Most of the strains listed in Table 2 are type strains, so we are confident that the named species in Table 2 are accurately classified. Among the 12 Lacticaseibacillus strains, L. zeae LI220 is a laboratory isolate, and L. casei KCTC 13086, L. paracasei MGB 0543, and L. rhamnosus KCTC 13088 is not type strains, but has been applied to several studies. Culture collection confirmed these species by 16S rRNA gene sequencing method or whole-genome sequencing. Among the strains listed in Table 2, we denoted the type strain as T.

Line 98: T, type strain

Table 2: We indicated the type strain.

  1. Line 94: But didn't the authors say above that 16S rRNA gene sequencing was not adequate to differentiate between strains in the L. casei group? Lines 46-47.

Response: 16S rRNA gene sequencing is a gold standard for bacterial identification. Although it has been used in many previous studies to identify bacteria, recent studies have reported that closely related species, including L. casei group, cannot be distinguished by the 16S rRNA gene. In this study, the 16S rRNA gene sequencing was used for comparison with our method and showed that our method was able to distinguish the L. casei group more accurately than 16S rRNA gene sequencing.

  1. Line 105: The methodology requires clarification. firstly, cultures were not used but apparently 20 ng of DNA extracted from cells of 9 species was mixed. Is this correct? If yes, then please write the details of what was actually done and do not refer to 'cultures' (cultures contain media plus cells), DNA was extracted from cells and mixed - not the cells obtained from cultures?

Response: In this study, cultures were not used, but 20 ng of DNA extracted from cells of nine species was mixed. As you recommended, we revised the sentence in lines 117-120 as follows:

Lines 117-120: pure and mixed DNA of nine species, including species mainly found in raw milk and closely related to L. zeae. For the preparation of mixed DNA, 20 ng of DNA was extracted from cells of nine species and mixed.

  1. Line 117: Table 2?

Response: As you recommended, we added the ‘Table 2’ in line 129 as follows:

Line 129: using 61 bacterial strains (Table 2).

  1. Line 121: Again, look at lines 46-47 - why use a tool which is not very precise?

Response: 16S rRNA gene sequencing is a gold standard for bacterial identification. Although it has been used in many previous studies to identify bacteria, recent studies have reported that closely related species, including L. casei group, cannot be distinguished by the 16S rRNA gene. In this study, the 16S rRNA gene sequencing was used for comparison with our method and showed that our method was able to distinguish the L. casei group more accurately than 16S rRNA gene sequencing.

  1. Line 125: No, the genome sequences are not 'misclassified': the names of the bacteria when first named may be inaccurate, particularly when many more genomes have been published to allow phylogenetic analysis. references 12 and 21 are earlier publications from this group, so this is inappropriate citation (citing this group's own opinion). There are many references that could be used to validate this statement e.g. Ghosh, as mentioned earlier.

Response: As you recommended, we corrected the citation for this sentence and revised the sentence in lines 142-143 as follows:

Lines 142-143: Previous studies have reported that public databases contain misnamed genomes for phenotypically closely related species [23–25].

  1. Line 142: This data assumes that the names of the strains are, indeed, the right species. The list of 124 strains had only 3 L. casei, including ATCC393 which was previously proposed to be 'L. zeae'. One strain, 12A, is a paracasei, so it is not too surprising that your own data has picked this up - with one 'L. casei' clustering with L. paracasei (in A and B) and one 'L. casei' clusters with L. zeae and chiayiensis in A (indicating that the core genome is similar) while two 'casei' cluter with zeae and chiayiensis in B (which means that the pan-genomes are similar). So aren't all of these strains (zeae, casei - 2 strains - and chiayiensis) very similar?

Response: In the previous study, L. zeae, L. casei, and L. chiayiensis were reported to be very similar (Huang et al., 2020, Int. J. Syst. Evol. Microbiol.), and the same result was observed in this study. We added the sentence in lines 153-156 as follows:

Lines 153-156: In the phylogenetic analysis, L. casei, L. chiayiensis, and L. zeae were very similar, which is similar to a previous study that three species were clustered adjacent to each other using a phylogenetic tree based on core genome MLST [5].

  1. Line 159: Yes, the similarity with L. rhamnosus strains is around 30%, and for enterococci proteins, and less for other species of LAB - including leuconostoc, lactococci. So specificity would then depend on the primer design outside the regions of sequence similarity. Was this the case?

Response: The specific gene of L. zeae showed within 36% homologies with other strains, such as Enterococcus, Lacticaseibacillus, and Lactococcus. However, the protein sequences did not match continuously with other strains. Also, the nucleotide sequence of glycosyltransferase shows less than 3.4% homologies with other strains. Therefore, the specificity did not change even when the primer design outside the regions of sequence similarity.

As you recommended, we newly performed an analysis demonstrating similarity between the glycosyltransferase in L. zeae and other strains and added the sentence and supplementary material in lines 86-89, 181-184, and 199-201 as follows:

Lines 86-89: The unique gene for L. zeae was compared with other strains through BLASTP search against the Kyoto Encyclopedia of Genes and Genomes (KEGG) database, and sequence alignment was performed using Clustal Omega (https://www.ebi.ac.uk/Tools/msa/clustalo/).

Lines 181-184: The glycosyltransferase of L. zeae was compared with other bacterial strains, and as a result, it showed the highest homology with Enterococcus gallinarum (36% identity) and less than 34% homology with other Lacticaseibacillus strains (Table S1).

Lines 199-201: As shown in Figure S1, since the sequences did not match consecutively, designing a primer at any position within this gene does not result in amplification in other bacteria.

Table S1: We newly added Table S1 for similarity for glycosyltransferase between L. zeae and other strains.

Figure S1: We newly added Figure S1 for sequence alignment for glycosyltransferase between L. zeae and other strains.

  1. Line 172: sequence alignment needs to be discussed here and data provided, not just stated like this. Perhaps in Supplementary Material?

Response: As you recommended, we newly performed an analysis demonstrating similarity between the glycosyltransferase in L. zeae and other strains and added the sentence and supplementary material in lines 86-89 and 199-201 as follows:

Lines 86-89: The unique gene for L. zeae was compared with other strains through BLASTP search against the Kyoto Encyclopedia of Genes and Genomes (KEGG) database, and sequence alignment was performed using Clustal Omega (https://www.ebi.ac.uk/Tools/msa/clustalo/).

Lines 199-201: As shown in Figure S1, since the sequences did not match consecutively, designing a primer at any position within this gene does not result in amplification in other bacteria.

Figure S1: We newly added Figure S1 for sequence alignment for glycosyltransferase between L. zeae and other strains.

  1. Line 185: How this was done is not clear from the description in the methods - see earlier 'bubble' to explain this statement.

Response: As you recommended, we revised the sentence in line 215 as follows:

Line 215: using pure and mixed DNA extracted from cells of nine species

  1. Line 193: Please discuss the limitations of the research: food samples (which may contain many species of LAB) were not tested and the presence of other LAB species which may contain similar genes was not tested.

Response: Thank you for your comment. As you recommended, we newly tested food sample artificially inoculated with L. zeae, and newly tested the presence of other LAB species which may contain similar genes. And, we added the sentence for the test results in lines 86-89, 127, 134-139, 181-184, 199-201, and 266-273 as follows:

Line 127: 2.4. Evaluation and application of real-time PCR

Lines 134-139: Our method was applied to spiked food sample. The milk sample was purchased from a local market in Korea and was confirmed to be free of L. zeae. A spiked milk sample was prepared according to a previous study [21,22]. Briefly, 25 ml milk sample was inoculated with L. zeae at a concentration of 103 to 109 CFU/ml. Genomic DNA was extracted inoculated sample according to the method described above, and real-time PCR was performed.

Lines 266-273: The quantification of genomic DNA in a food sample was conducted by artificially adding L. zeae strain to milk. The real-time PCR method developed in this study could successfully be used to identify L. zeae at a concentration of 103 to 109 CFU/ml in milk (Figure 6). Samples artificially inoculated with L. zeae (Ct values: 14.97 to 34.78) had a slightly higher Ct value than the pure culture of L. zeae strain, which seemed to slightly affect the efficiency of real-time PCR. This may be due to the presence of PCR inhibitors from fast and protein in food [21]. Therefore, our real-time PCR method was able to identify L. zeae in the food matrix.

Figure 6: We newly added the result for the detection of L. zeae in spiked milk sample in Figure 6.

Lines 86-89: The unique gene for L. zeae was compared with other strains through BLASTP search against the Kyoto Encyclopedia of Genes and Genomes (KEGG) database, and sequence alignment was performed using Clustal Omega (https://www.ebi.ac.uk/Tools/msa/clustalo/).

Lines 181-184: The glycosyltransferase of L. zeae was compared with other bacterial strains, and as a result, it showed the highest homology with Enterococcus gallinarum (36% identity) and less than 34% homology with other Lacticaseibacillus strains (Table S1).

Lines 199-201: As shown in Figure S1, since the sequences did not match consecutively, designing a primer at any position within this gene does not result in amplification in other bacteria.

Table S1: We newly added Table S1 for similarity for glycosyltransferase between L. zeae and other strains.

Figure S1: We newly added Figure S1 for sequence alignment for glycosyltransferase between L. zeae and other strains.

  1. Line 199: was DNA extracted from an artificial mixture of cells or by adding 20 ng of DNA extracted from pure cultures? This is not clear from this description or from the methods section.

Response: As you recommended, we revised the sentence in lines 229-230 as follows:

Lines 229-230: mixed DNA prepared by mixing 20 ng of DNA extracted from pure cultures.

  1. Line 223: The difficulty is that L. zeae was the name proposed for ATCC393 and the phylograms provided show how similar zeae and 'casei' are. More strains of 'casei' validated as similar to ATCC393 (NCBI BLAST would identify these) should be used to demonstrate specificity, rather than showing that the primer work with zeae.

Response: As you recommended, we reanalyzed the pan-genome by adding the 17 genomes of L. casei (total 20 genomes) and demonstrated specificity. The added genome information and reanalysis results were added to the in lines 13, 15, 65, 67, 74, 159, 165, 167, 167-168, 170-171, 174 as follows:

Line 13, 67, 167: 141 genomes

Line 65, 74, 159, 165: 141 genome sequences

Line 13: yielded an 17,978 pan-genome

Line 15: 136 other genomes

Lines 167-168: The 17,978 pan-genome obtained from 141 genomes is composed of 144 core genes and 4,271 unique genes

Lines 170-171: biogenesis (38.6%), and nucleotide transport & metabolism (6.7%).

Line 174: Among the 13,563 accessory genes

Table 1: We added genome features of the newly 17 L. casei in Table 1.

Figure 1: We added pan-genome distribution for L. casei.

Figure 2: We added phylogeny analysis based on core-genome and pan-genome for L. casei.

  1. Line 226: How many of these were genuine L. casei, given that the nomenclature is still confusing and mis-named strains are in the literature and culture collections.

Response: Most of the strains listed in Table 2 are type strains, so we are confident that the named species in Table 2 are accurately classified. Among the 12 Lacticaseibacillus strains, L. zeae LI220 is a laboratory isolate, and L. casei KCTC 13086, L. paracasei MGB 0543, and L. rhamnosus KCTC 13088 is not type strains, but has been applied to several studies. Culture collection confirmed these species by 16S rRNA gene sequencing method or whole-genome sequencing. Among the strains listed in Table 2, we denoted the type strain as T.

Line 98: T, type strain

Table 2: We indicated the type strain.

Reviewer 2 Report

Very interesting manuscript, that I enjoyed reading due to its degree of novelty and to the well-built argumentation. However, English needs a thorough revision (very poor grammar and phrase structure throughout most of the manuscript). Another point that needs to be addressed is the low number of Lacticaseibacillus species represented in this study (5 out of 26 – less than 20%). This shortcoming – which we agree is not easy to circumvent, given that species that have been described very recently might not have easily accessible isolates – should be discussed in text, so that the readers have a better picture of the scope of the authors’ findings.

Detailed comments follow:

Lines 26 to 28 - This part of the text is not clear. It seems to imply that the Lacticaseibacillus genus consists of 4 species only, whereas at present  26 species are on the List of Prokaryote Names with Standing in Nomenclature for this species (L. absianus, L. baoqingensis, L. brantae, L. camelliae, L. casei, L. chiayiensis, L. daqingensis, L. hegangensis, L. hulanensis, L. jixinaensis, L. manhotivorans, L. minshuiensis, L. nasuensis, L. pantheris, L. paracasei, L. porcinae, L. rhamnosus, L.  saniviri, L. sharpae, L. songhuajiangensis, L. suibinensis, L. thailandensis, L. yichunensis, L. zeae, and L. zhaodongensis. The text should be modified in order to inform the reader that the species mentioned are only 4 out of the 26 species in the genus. This is   and important datum for the reader to assess how broad was the spectrum of this study.

Line 27 – Please replace “species” with “genus”, for the sake of clarity.

Line 28 - For the sake of language clarity, consider replacing "taxonomic relationship" with "phylogeny" or "taxonomic position".

Line 61 - By Lactobacillus-related species do you mean that these species are related to the species in the Lacticaseibacillus genus? This sentence should be rewritten to improve clarity.

Lines 63-64 - With genomes of only 5 out of the 26 species in the genus, it is doubtful whether the chosen gene would really be of use to distinguish L. zeae from Lacticaseibacillus species other than L. casei, L. paracasei, and L. chyaiensis. Your work is, nevertheless, relevant since these are the main species found in milk and milk-related environments. Your text should, however, make this limitation clear for the readers.  

Table 1 - Some of the names of bacterial species are mentioned in this table for the first time and should, therefore, be given in full.

Table 2 - Give in full just the bacterial names that were not mentioned previously.

Line 104 – Conditions instead of condition

Lines 125 – 126 - Example of a sentence that needs rewriting to improve its structure (use of English).

Line 144 - The pangenome ... is composed. Please replace are with is.

Lines 148 – 151 - Very interesting results. Sentence structure, however, needs to be improved.

Line 154 - acidilactici. Please correct.

Line 153 - Sentence structure; please have the whole text revised by a native English speaker familiarized with scientific writing. I pointed out only a few examples of sentences that need rewriting due to poor use of English, but most of the text needs to be improved.

Lines 155-156 - Another example of poor sentence structure and grammar.

Lines 221-223 - The authors correctly identified one of the main limitations of this study and made it clear for prospective readers. However, in my opinion, the low number of species of the Lacticaseibacillus genus in this study is another relevant limitation that remains to be discussed. Moreover, the authors worked with OTUs that are mainly relevant for milk and related environments. Clearly, the difficulty to find isolates for the most recently added species is a factor that hinders this type of study and justifies the choice of bacterial species included in the present study. However, the authors do not discuss this aspect, which is of relevance for the potential readers of this manuscript.

Author Response

Response to Reviewer 2 Comments

Very interesting manuscript, that I enjoyed reading due to its degree of novelty and to the well-built argumentation. However, English needs a thorough revision (very poor grammar and phrase structure throughout most of the manuscript). Another point that needs to be addressed is the low number of Lacticaseibacillus species represented in this study (5 out of 26 – less than 20%). This shortcoming – which we agree is not easy to circumvent, given that species that have been described very recently might not have easily accessible isolates – should be discussed in text, so that the readers have a better picture of the scope of the authors’ findings.

Detailed comments follow:

  1. Lines 26 to 28 - This part of the text is not clear. It seems to imply that the Lacticaseibacillus genus consists of 4 species only, whereas at present 26 species are on the List of Prokaryote Names with Standing in Nomenclature for this species (L. absianus, L. baoqingensis, L. brantae, L. camelliae, L. casei, L. chiayiensis, L. daqingensis, L. hegangensis, L. hulanensis, L. jixinaensis, L. manhotivorans, L. minshuiensis, L. nasuensis, L. pantheris, L. paracasei, L. porcinae, L. rhamnosus, L. saniviri, L. sharpae, L. songhuajiangensis, L. suibinensis, L. thailandensis, L. yichunensis, L. zeae, and L. zhaodongensis. The text should be modified in order to inform the reader that the species mentioned are only 4 out of the 26 species in the genus. This is and important datum for the reader to assess how broad was the spectrum of this study.

Response: As you recommended, we revised the sentence in lines 26-32 as follows:

Lines 26-32: The genus Lacticaseibacillus consists of 26 species (L. absianus, L. baoqingensis, L. brantae, L. camelliae, L. casei, L. chiayiensis, L. daqingensis, L. hegangensis, L. hulanensis, L. jixianensis, L. manihotivorans, L. mingshuiensis, L. nasuensis, L. pantheris, L. paracasei, L. porcinae, L. rhamnosus, L. saniviri, L. sharpeae, L. songhuajiangensis, L. suibinensis, L. suilingensis, L. thailandensis, L. yichunensis, L. zeae, and L. zhaodongensis), and Lacticaseibacillus zeae is one of the members of the Lacticaseibacillus genus, along with L. casei, L. paracasei, L. rhamnosus, and L. chiayiensis.

  1. Line 27 – Please replace “species” with “genus”, for the sake of clarity.

Response: As you recommended, we replaced “species” with “genus” in lines 31-32 as follows:

Lines 31-32: the Lacticaseibacillus genus

  1. Line 28 - For the sake of language clarity, consider replacing "taxonomic relationship" with "phylogeny" or "taxonomic position".

Response: As you recommended, we replaced “taxonomic relationship” with “taxonomic position” in line 33 as follows:

Line 33: the taxonomic position of L. zeae

  1. Line 61 - By Lactobacillus-related species do you mean that these species are related to the species in the Lacticaseibacillus genus? This sentence should be rewritten to improve clarity.

Response: As you recommended, we revised the sentence in line ## as follow:

Line 65: A total of 141 genome sequences representing nine lactic acid bacterial species were

  1. Lines 63-64 - With genomes of only 5 out of the 26 species in the genus, it is doubtful whether the chosen gene would really be of use to distinguish L. zeae from Lacticaseibacillus species other than L. casei, L. paracasei, and L. chyaiensis. Your work is, nevertheless, relevant since these are the main species found in milk and milk-related environments. Your text should, however, make this limitation clear for the readers.

Response: As you recommended, we added the sentence in lines 67-69 as follows:

Lines 67-69: To overcome the limitation that 141 genomes contained only 5 out 26 species in the genus Lacticaseibacillus, nine species isolated from raw milk, the main habitat of L. zeae, were included in the genome analysis.

  1. Table 1 - Some of the names of bacterial species are mentioned in this table for the first time and should, therefore, be given in full.

Response: As you recommended, we listed the full names of the first mentioned bacterial species in Table 1.

  1. Table 2 - Give in full just the bacterial names that were not mentioned previously.

Response: As you recommended, we only gave full names that not previously mentioned bacteria in Table 2.

  1. Line 104 – Conditions instead of condition

Response: As you recommended, we changed condition to conditions in line 116 as follows:

Line 116: Real-time PCR conditions

  1. Lines 125 – 126 - Example of a sentence that needs rewriting to improve its structure (use of English).

Response: As you recommended, we revised the sentence in lines 142-143 as follows:

Lines 142-143: Previous studies have reported that public databases contain misnamed genomes for phenotypically closely related species

  1. Line 144 - The pangenome ... is composed. Please replace are with is.

Response: As you recommended, we changed are to is in line 167 as follows:

Line 167: genomes is composed

  1. Lines 148 – 151 - Very interesting results. Sentence structure, however, needs to be improved.

Response: As you recommended, we revised the sentence in lines 171-173 as follows:

Lines 171-173: Also, the genes common to five genome sequences of L. zeae were classified to COG categories, mainly functioning in the defense system (16.7%), amino acid transport and metabolism (11.1%), and cell wall/membrane/envelop biogenesis (11.1%).

  1. Line 154 - acidilactici. Please correct.

Response: As you recommended, we corrected the species name in line 177 as follows:

Line 177: P. acidilactici

  1. Line 153 - Sentence structure; please have the whole text revised by a native English speaker familiarized with scientific writing. I pointed out only a few examples of sentences that need rewriting due to poor use of English, but most of the text needs to be improved.

Response: As you recommended, we revised the sentence in lines 175-178 as follows:

Lines 175-178: The 18 genes were aligned with 72,899,005 bacterial sequences through blast analysis. As a result, three genes existed in other microorganisms such as Enterococcus durans, Pediococcus damnosus, and P. acidilactici, and the remaining 15 genes were specifically present in L. zeae.

  1. Lines 155-156 - Another example of poor sentence structure and grammar.

Response: As you recommended, we revised the sentence in lines 179-181 as follows:

Lines 179-181: Among these, the gene specific to L. zeae was finally selected as glycosyltransferase family 8 (accession no. KRK10099.1) in consideration of the GC content and length.

  1. Line 173: Again, poor grammatical structure. Please rewrite - and do a thorough check of the manuscript to improve the use of English.

Response: As you recommended, we revised the sentence in lines 201-202 as follows:

Lines 201-202: Therefore, we confirmed that this gene was specifically present in L. zeae.

  1. Lines 221-223 - The authors correctly identified one of the main limitations of this study and made it clear for prospective readers. However, in my opinion, the low number of species of the Lacticaseibacillus genus in this study is another relevant limitation that remains to be discussed. Moreover, the authors worked with OTUs that are mainly relevant for milk and related environments. Clearly, the difficulty to find isolates for the most recently added species is a factor that hinders this type of study and justifies the choice of bacterial species included in the present study. However, the authors do not discuss this aspect, which is of relevance for the potential readers of this manuscript.

Response: As you recommended, we added the sentence for the main limitation of this study in lines 254-256 as follows:

Lines 254-256: Also, our data has a shortcoming in that a low number of Lacticaseibacillus species (5 out of 26 species) represented in this study because species that have been described very recently have not been easily accessible isolates

Round 2

Reviewer 1 Report

The authors have made an excellent effort in address the many comments made during review of the original manuscript.  They have added additional L. casei strains for comparative purposes, provided additional analyses (in the main text and in supplementary information) and have bettered argued their case for use of the PCR method described therein for detecting L. zeae in food (the example given is milk seeded with a known strain, so demonstrating that the method can be used in practice). One small point:  lactobacilli is always spelt with lower case 'L' so please remove the capital for this word (picked up once in the rejoinder).  A second small point is the description of using 20 ng of DNA in a mixture:  the current phrasing in the methods is still not quite accurate.  DNA was extracted from the cells of XXX species and 20 ng of each mixed to provide a template for PCR amplification.   Please look at the phrasing again to see if the above is captured.

Author Response

Response: Thank you for your comment. As you recommended, we changed Lactobacilli to lactobacilli and revised the sentence of using 20 ng of DNA in a mixture in lines 81, 104, 109, 118-120, and 229-230 as follows:

Line 81, 104, 109: lactobacilli

Lines 118-120: For the preparation of mixed DNA, DNA was extracted from the cells of nine species and 20 ng of each mixed to provide a template for PCR amplification.

Lines 229-230: (D) melting curve obtained from the mixed DNA prepared by extracting DNA from the cells of nine species and mixing 25 ng each.